# Habituation Protocols Improve Behavioral and Physiological Responses of Beef Cattle Exposed to Students in an Animal Handling Class

**DOI:** 10.3390/ani11082159

**Published:** 2021-07-21

**Authors:** Aska Ujita, Zachary Seekford, Michelle Kott, Guillermo Goncherenko, Nicholas W. Dias, Erica Feuerbacher, Luciana Bergamasco, Leonie Jacobs, Dan E. Eversole, João A. Negrão, Vitor R. G. Mercadante

**Affiliations:** 1Basic Science Department, Faculty of Animal Science and Food Engineering, University of São Paulo, Pirassununga, SP 13635-900, Brazil; aska_ujita@hotmail.com (A.U.); jnegrao@usp.br (J.A.N.); 2Department of Animal and Poultry Sciences, College of Agriculture and Life Sciences, Virginia Tech, Blacksburg, VA 24060, USA; zseekford@ufl.edu (Z.S.); mkott@vt.edu (M.K.); nichw16@vt.edu (N.W.D.); enf007@vt.edu (E.F.); lucianab@vt.edu (L.B.); jacobsl@vt.edu (L.J.); deversol@vt.edu (D.E.E.); 3Polo Agroflorestal CENUR-Noroeste, Faculdad de Veterinaria, Universidad de la Republica, Melo, CL 91500, Uruguay; guillermogoncherenko@gmail.com; 4Department of Large Animal Clinical Sciences, VA-MD College of Veterinary Medicine, Virginia Tech. Blacksburg, VA 24060, USA

**Keywords:** beef cattle, behavior, habituation, human handling, cortisol, positive stimulation

## Abstract

**Simple Summary:**

Students in agricultural programs have the opportunity to interact with animals during different teaching activities. However, students’ interactions with livestock may be distressing to the animals and can affect the students’ and animals’ safety. We investigate whether two human-animal habituation strategies, one with positive tactile stimulation and one without, would improve behavioral and physiological responses of beef heifers during a livestock handling class. Overall, heifers that received a habituation treatment had more positive behavior responses, and decreased physiological stress responses in comparison to heifers that were not exposed to habituation. Furthermore, the heifers exposed to the habituation with a positive tactile stimulation had the greatest improvements in behavior in comparison to control and non-stimulated heifers, exhibiting more positive behaviors when interacting with humans. Strategies to habituate cattle to human interaction with positive stimulation aligned with training humans that handle and interact with cattle on best practices and cattle behavior can improve behavior, reduce stress-related physiological responses and enhance safety for both humans and animals.

**Abstract:**

Our objective was to determine the impact of different habituation protocols on beef cattle behavior, physiology, and temperament in response to human handling. Beef heifers were exposed to three habituation strategies: (1) tactile stimulation (brushing) in the working chute for seven consecutive days (STI; *n* = 18); (2) passage through the working chute for seven consecutive days (CHU; *n* = 19) and; (3) no habituation (CON; *n* = 19). Individual heifer respiratory rate (RR; n/min), internal vaginal temperature (VAGT; °C), and blood cortisol were measured. Further, behavior parameters were observed to generate a behavior score, and heifer interaction with students and their behavioral responses were recorded. Habituation with STI and CHU resulted in improved numerical behavioral scores compared to CON, and greater (*p* ≤ 0.05) handling latencies. Vaginal temperature was decreased in STI compared to CHU and CONT (*p* ≤ 0.05). Cortisol concentration did not differ among treatments, but decreased (*p* ≤ 0.05) from the start of the experiment to 14 days after treatment initiation. Both habituation protocols showed benefits, but heifers that received the positive tactile stimulation in the chute had the greatest behavior improvements. Furthermore, these heifers responded more calmly during student-animal interactions in class, which is beneficial for the students’ and animals’ safety.

## 1. Introduction

Cattle behavior is the result of an interaction between genetic and environmental factors [1], thus it is important to develop efficient handling techniques that reinforce positive cattle behavior and low reactivity to humans. Accordingly, the adoption of positive handling practices may prevent the animal from associating humans with a negative experience, especially when introduced at young age. Animals that experience a positive human-animal interaction earlier in life are prone to be less reactive and more accustomed to human handling [2]. Thus, habituating young animals could result in a long-term positive human-animal relationship [3,4,5].

Human-animal interaction is a major factor when dealing with animals, frequently observed during any kind of handling. In addition to an agricultural setting, cattle are used in educational institutions, such as universities with agricultural programs. These educational institutions may use cattle for experiential learning and as a source of income. Experiential learning activities are important to strengthen students’ cognitive learning as well as development and practice of animal handling techniques [6,7].

Students in agricultural programs have the opportunity to interact with animals during different teaching activities. However, students’ interactions with livestock, including cattle may be distressing and difficult due to students’ lack of experience and knowledge of animal behavior, and infrequent exposure and interaction of humans and animals to each other. Therefore, student-herd interactions may result in negative and dangerous experiences for both humans and animals. Cattle may respond with undesirable or aversive behaviors when faced with a new or challenging situation [5,8,9], such as student interactions. These negative interactions may lead to increased labor costs, injuries or fatal accidents [10], as well as decreased profits due to reduced performance and poor product quality.

Novelty it is a potentially stressful stimulus that can impact animals negatively [11]. Cattle that are infrequently handled, such as in teaching herds, show both behavioral and physiological stress responses to a handler. Previous research showed that a combination of brushing or stroking, a conversation, and affectionate gestures from humans directed to the animals results in positive animal behavior and also easier handling [2,3,4,12,13,14,15]. Such strategies could also be applied at educational institutions, safeguarding both animal productivity and animal welfare. However, this has not yet been researched in an educational context, with students interacting with young cattle.

The study objective was to determine the impact of different habituation protocols on beef cattle behavior, physiology, and temperament in response to chute handling and human interaction. We hypothesized that European (*Bos taurus taurus*) beef heifers in a teaching herd that received the positive tactile stimulation as a habituation to chute handling and human interactions during a student halter break class, would show more docile behavior when interacting with students (handling, touching and approach), compared to those heifers without habituation or those only habituated to the chute without tactile stimulation.

## 2. Materials and Methods

The study (Institutional Animal Care and Use Committee (IACUC) number 19-215), was conducted at the Virginia Tech Beef Cattle Center in Blacksburg (VA, USA), cattle of Angus, Simental, Hereford and Charolais breeds were enrolled in the study. Prior to the initiation of the study, all animals were maintained on a single pasture and fed *ad libitum* with silage and hay and had limited human interaction, except in standard farm managements: such as parturition, and pre-weaning weighing and vaccination.

### 2.1. Experimental Treatments

Cattle enrolled in this study were part of a halter breaking and cattle handling class. Three weeks prior to the start of the class (day −21), 56 heifers from the university teaching herd (12 ± 2 months of age and 398.70 ± 5.08 kg body weight) were first scored in the chute according to the methods described in Table 1 and the individual measures were summed into a final score. These final scores were classified as high, medium and low reactivity, and cattle from each reactivity class were then randomly assigned to one of three human-animal interaction treatments: (1) habituation group with tactile stimulation (brushing) in the working chute (Stimulated group—STI, *n* = 18) during seven consecutive days; (2) habituation group with passage through the working chute (Chute group—CHU, *n* = 19) during seven consecutive days and; (3) a control group with no habituation, which represents the standard procedure for the teaching herd (Control group—CON, *n* = 19).

Figure 1 shows a timeline of the trial. At day −21, all animals were scored for reactivity classification (Table 1). Habituation treatments took place starting 14 days prior to the scheduled student-animal interactions (halter breaking practice class: HBPC, Day 0).

#### 2.1.1. Tactile Stimulation (STI Group)

The tactile stimulation treatment was repeated for seven consecutive days (day −14 to day −8), applied once a day starting at 8:00 AM, for 10 min per animal. Heifers were guided to the working chute and were held in between the squeeze chute and the crowd box stall (Figure 2), with the sliding gates closed. Four animals were habituated simultaneously by four trained people. The tactile stimulation consisted of brushing and touching the animal with a long handle broom, short handle broom, and hands one method at a time and equally distributed during the 10 min period. After all the STI animals received their stimulation, the herd was returned to their respective pasture.

#### 2.1.2. Passage through the Chute (CHU Group)

The chute treatment consisted of heifers being habituated to the squeeze chute (Figure 2). The cattle were guided to the chute and were allowed to walk through the working and squeeze chutes without restraint or any human interaction, taking less than one minute per animal. The herd were moved as a group in the working facility, but individually moved through the working chute and squeeze chute. After all CHU animals completed their habituation, the herd was returned to their respective pasture. This activity occurred on the same days as the STI group habituation and was repeated for seven consecutive days (day −14 to day −8). Chronologically, CHU animals received their treatments after the STI group on all 7 days.

#### 2.1.3. Control (CON Group)

The animals from the CON treatment were not handled until the halter breaking practice class (HBPC) on day 0. In other words, they were kept on pasture throughout the habituation period of STI and CHU groups (day −14 to day −8) and did not undergo any human-animal interactions that could be considered habituation to handling or to the chute.

#### 2.1.4. Halter Breaking Practice Class (HBPC)

Forty-nine students were enrolled in the “Livestock Management and Handling Laboratory” class (APSC 2114) with the objective of learning through hands-on activities safety in livestock handling, animal behavior, care, housing, and managerial practices related to beef cattle, sheep and swine. Students provided prior consent to participate in the study and be recorded (Institutional Review Board (IRB) protocol number 19-933).

For the HBPC, each student needs to work with one animal. Students were randomly paired with one of the initial 56 heifers enrolled in the study resulting in 17 heifer-student pairs from CHU, and 16 pairs from both STI and CON treatments.

Animal behavioral responses to student handling were collated from video footage recorded with monitoring cameras (IP Bullet camera FLPB133F, FLIR Systems Inc., OR, USA). Human behavior was not scored but rather the animals’ reactivity to the handling. Measurements (Table 2) were taken during four stages of student-animal interactions:Stage 1: the student brought the animal through the working chute and into the squeeze chute for rope halter placement, which was performed by the student.Stage 2: the student trailed the animal from the squeeze chute to the respective box stall (Figure 1), allowing the animal to move freely (dragging the halter‘s rope without any tension from the student).Stage 3: the student tied the animal to the fence inside a box stall as part of halter habituation. The student did not interact with the heifer for 5 min. Thereafter, the student scratched the animal with a long wooden stick with a small hook at the tip (often called a show stick). The scratching continued during the remainder of the class, but only the first two minutes were evaluated.Stage 4: After being haltered for approximately 30 min, the student led the animal to exit the box stall and enter the pasture.

### 2.2. Experimental Measurements

All data were collected in the squeeze chute in the mornings on days −21 and 14 (Figure 2).

#### 2.2.1. Respiratory Rate and Vaginal Internal Temperature

Respiratory rate (RR; n/min) and internal vaginal temperature (VAGT; °C) were collected from animals while they were restrained in the squeeze chute. The RR was measured by counting the flank breathing movements for 15 s, then multiplying it by four to obtain the respiratory rate per minute. The VAGT was measured by placing a conventional thermometer (Digital Thermometer KD-1760, 180 INNOVATIONS, Denver, CO, USA) in the heifer’s vagina until the temperature stabilized. For the STI treatment group, these measurements were collected in the squeeze chute right after tactile stimulation in the working chute.

#### 2.2.2. Cortisol Concentration in Blood Plasma

Immediately as the heifer entered the squeeze chute and was restrained blood samples were taken on days −21 and 14 from the coccygeal vein using vacuum collecting tubes (Vacutainer^®^, 10 mL; Becton Dickinson, Franklin Lakes, NJ, USA) containing sodium heparin (148 USP units), placed immediately on ice for < 4 h, and centrifuged at 2400× *g* for 15 min at room temperature. Plasma was collected and stored at −20 °C until further analysis. Plasma cortisol concentrations were determined in duplicates by using a chemiluminescence assay (Immulite 2000 Xpi, Siemens Medical Solutions, Princeton, NJ, USA) with an intra- and inter-assay CV of 5.3% and 4.8%, respectively.

#### 2.2.3. Behavioral Parameters in the Working Chute

Behavioral parameters are defined in Table 1 and were assessed through direct observation from the moment heifers entered the working chute until they exited the squeeze chute (Figure 1) on days −21 and 14. Balking during squeeze chute entry (BALK_ENT), velocity score to enter the squeeze chute (VEL_ENT) were determined as well as locomotion from working chute to squeeze chute and exit (LOC) and squeeze chute exit behavior as balking (BALK_EXT), velocity score to enter the squeeze chute (VEL_EXT) and latency to enter squeeze chute (LAT_ENT). On days of blood collection in the squeeze chute, behavior score was evaluated during restraint (BC).

Flight time (FT, sec/m) was measured as time taken to pass two pairs of infrared sensors (Wireless Electric Eyes, FarmTek Inc., North Wylie, TX, USA, Figure 3). The flight time was automatically recorded for each heifer as it exited the squeeze chute [16]. The latency to enter the working chute (sec) was recorded from videos taken, using a predetermined point of reference as starting point located at the first sliding gate after the crowd pen.

#### 2.2.4. Behavioral Measurements in the Holding Pen

Pen score (PS) methodology was adapted [17] and executed as shown in Table 3. After exiting the squeeze chute, the heifer was brought to a box stall individually, and scored for reactivity to a human presence and approach. In the first minute, the human would stand still and assign score 1 if the heifer approached voluntarily (Table 3). Thereafter, the human approached the heifer calmly, moving at approximately 0.7 m/sec, to assign the other behavioral scores (2 to 7). The procedure was executed in pairs: one person inside the pen, and another outside the box stall assigning the scores. All animals were scored by the same two observers.

### 2.3. Statistical Analysis

Data analysis was performed using SAS 9.4 (SAS Institute, INC., Cary, NC, USA). We used a randomized block design. The treatments (STI, CHU, CON), sampling day (days −21 and 14) and their interaction were used as fixed effects. The data were evaluated as repeated measures over time (sampling day). The covariance matrices were tested for each trait and the structure showing the best fit was used [18]. Each animal was considered the experimental unit. When differences between treatments were found, a comparison of means by the t-test was performed, with a significance of 5%. No breed effect was found, but this factor was maintained in the model as a random effect for the best representation of the experimental design. The delta values (Δ) of behavioral parameters, RR, VAGT and cortisol over time were calculated as an additional response variable and were analyzed the same as the other response variables. The results were organized in two sections: a comparison between treatments (STI vs CHU vs CON groups) before and after habituation protocols (days −21 and 14) to determine the treatment effect on behavioral and physiological parameters over time; and a comparison between treatments (STI vs CHU vs CON groups) to determine the habituation effect on behavioral parameters during HBPC.

## 3. Results

### 3.1. Behavioral Indicators

The effect of treatment (STI, CHU, CON), day of sampling and their interaction on behavioral and physiological parameters are presented in Table 4.

Habituation with tactile stimulation (STI) resulted in improved numerical behavioral scores (BALK_ENT, LAT_ENT, VEL_ENT, LOC, BALK_EXT, FT and PS) compared to the habituation without tactile stimulation (CHU) but only LAT_ENT and BC were significant (*p* ≤ 0.05). Control heifers had worse (greater) values compared to habituated heifers on VEL_ENT, LOC, BC, VEL_EXT, and PS. Latency to enter the working chute (LAT_ENT) and reactivity during BC differed among habituation treatment groups (*p* ≤ 0.05), with shorter latencies in the CON treatment compared to STI and CHU, and greater scores for BC compared to STI and CHU (Table 4).

Reactivity improved on day 14 compared to day −21 (BALK_ENT, VEL_ENT, BC, BALK_EXT, VEL_EXT and PS; *p* ≤ 0.05). In contrast, LAT_ENT was longer on day 14 compared to day −21 (*p* ≤ 0.05).

The interaction between treatments and sampling day showed a significant effect on BALK_ENT and VEL_EXT (*p* ≤ 0.05) (Table 4; Figure 4A,C), starting with similar values on day −21 (Figure 4) and showing a significant decrease on day 14, except for the VEL_EXT duration of the CON group (Figure 4C). The delta values (Figure 4D,F) showed that the STI group had the greatest decrease compared to CHU and CON groups, with lesser values (estimates and delta) and significant difference (*p* ≤ 0.05), except on exit velocity, which differed only from CON group (*p* ≤ 0.05).

#### Physiological Parameters

Treatments affected VAGT (*p* ≤ 0.05), with reduced temperatures in the STI group compared to CHU and CON (Table 4). There was an effect of day, but not habituation treatment, on VAGT, RR and CORT with lesser values on day 14 compared to day −21 (Table 4; *p* ≤ 0.05). VAGT was affected by treatment depending on sampling day (*p* ≤ 0.05; Table 4), with treatments groups showing similar temperatures on day −21 (Figure 4B) and a greater temperature decrease in STI compared to CHU and CON on day 14 (Figure 4F).

### 3.2. Student-Animal Interaction during HBPC

The STI treatment resulted in lesser values compared to the CON treatment for pen exit velocity and halter scores (*p* ≤ 0.05; Table 5). STI and CHU values for those variables did not differ. CHU heifers’ exit velocity was slower compared to CON heifers (Table 5). Other behavioral responses during class did not differ among habituation treatments (Table 5).

## 4. Discussion

Although habituation effect was significant only in three (LAT_ENT, VAGT and BC) of all traits assessed, these results indicate that seven days of habituation (STI and CHU) is a useful tool to decrease reactivity of heifers and increase ease of handling in working and squeeze chutes. This is in agreement with results from others [2] that also indicated seven days of habituation would improve the ease of cattle movement through a handling facility and squeeze chute. Thus, our results reinforce previous findings showing the beneficial impact of positive reinforcement (STI) or just habituation to handling and facilities (CHU) on animal reactivity [2,12,23].

Latency to enter the working chute can directly influence and be influenced by balking and velocity score to enter the working chute. An animal that balks (undesirable behavior) will have greater latency, but an exacerbated velocity to enter (also undesirable behavior) will decrease latency. The greater decrease in velocity in STI treatment indicates that brushing was more effective in improving this trait and heifers were calmer on day 14 in comparison to day 1. Our findings are similar to others where 36% of beef cattle showed slower velocities (walking rather than running) after 14 days of chute habituation with brushing [2], indicating a reduced fear of handling and improved entry behavior. Furthermore, in other species latency to approach or move away are associated with temperament during handling and exposure to handling facilities [24,25]. Therefore, it is important to associate latency with velocity and balking measures, since together these measurements better indicate desirable and undesirable behaviors, such as trust or fear that may prolong or not latency.

Although LOC did not differ, results show that CON heifers were more excited than habituated heifers. Few studies working with novelty tests affirm that fear motivates action in behavior such as locomotion [26,27,28]. In addition, previous research supports that a passive animal would show silence and inactivity to confront an unknown or uncomfortable situation, while active animals would be rely on fighting, struggling, vocalization and locomotion [27,29,30]. In our study, no passive heifers were identified and observed, suggesting that the herd is characterized by active or proactive animals. Thus, the exited locomotion of CON heifers indicates that this group was likely uncomfortable and stressed with handling, which may be the reason for the numerically increased RR. Stress is associated with physiological changes and respiratory rate is used to study aspects of stress and cognition [2,31,32,33]. Following this reasoning, we expected VAGT to be greater in CON heifers, but surprisingly CHU heifers had greater VAGT compared to other treatments. Nonetheless, vaginal temperature can vary during the different stages of the estrous cycle [34,35] and we did not evaluate puberty status or followed the estrous cycle of the heifers enrolled in this study which could have affected our results.

Concentration of cortisol was increased in heifers without habituation. Cortisol production and release is often increased in response to stress, being a predictive of anxiety-related behaviors and its concentration is relevant for animal welfare [36]. Other studies found that positive handling and habituation can reduce concentration of cortisol [4,37]. However, exact thresholds for concentration of cortisol to determine different levels of stress do not exist in cattle. Reports in the literature vary significantly, one study observed cortisol concentrations of 4.97 μg/dL in a control group and 2.94 μg/dL in the group exposed to habituation [4] while others reported cortisol concentrations of 0.70 μg/dL in non-stressed and 2.95 μg/dL in stressed groups [36]. Differences in environment, breed, age, animal category and even method of sampling and analysis need to be considered when comparing and contrasting cortisol results.

Flight time, flight speed and pen score are commonly used to assess behavior and temperament in cattle. The stimulated heifers in our study tended to have a decreased flight time, which is a desirable behavior and indicative of reduce stress and fear. Pen score was similar among treatments, however it decreased between day −21 and day 14 independent of treatment, indicating that all heifers improved their behavior when interacting with a human inside of small space. Previous research with lactating dairy cows indicates that spending even a short-time stroking the cows during milking can positively affect their response to humans, reducing avoidance distances of the animals at the feeding place, the barn and in the milking parlor [38]. Another study found that gently touching young pre-weaning beef calves resulted in lesser avoidance distance, greater voluntary approaches and fewer backward movements inside the stunning box at slaughter at 10 months of age [4]. Clearly positive interactions such as stroking, brushing and talking with a gentle voice are necessary to maintain a positive human-animal relationship with improved trust and reduced fear from both sides [15].

In the case of our study, seven days of habituation (with or without tactile stimulation) might not be enough to present a statistically significant differences in all traits observed, but could still improve the general behavior of the herd. Even a short interactions and handling events separated by long periods of time in between can improve cattle habituation to handling. The overall reduction in PS and cortisol concentration between day −21 and 14, independent of treatment highlights the importance of exposing young cattle to handling events and facilities. These findings are corroborated by other studies. Temperament scores decrease as the number of times an animal goes through a working facility increased [2,39,40].

There was treatment by day interaction for three traits assessed (BALK_ENT, VAGT and VEL_EXT) and the delta values, which represents the difference in the variables between day −21 and day 14, also differed. No difference in behavior and temperament assessment was detected at the beginning of the experiment among treatments. These results suggest that tactile stimulation is an alternative to facilitate handling and improve animal behavior, which might be linked to a decreased fear of human or facilities as other studies suggest [2,3,4,15,41]

### Student-Animal Interaction during HBPC

The HALTER trait represents the reactivity score when the student interacts with the heifer trying to place the halter in the heifer at the squeeze chute. Both habituation treatments resulted in decreased reactivity and increased ease to halter, suggesting that habituation is also a tool to reduce fear of human approach and handling. The PULL_N, that consists in how many times the animal pulled the rope during the first five minutes after being tied, did not differ statistically. However, STI heifers had a reduction in pull of approximately 50% in comparison to the other treatments.

Livestock involves a large variety of activities such as feeding, reallocation of animals (by moving from field to field or by loading animals on trucks/trailers), artificial insemination, roof care, dehorning, roping animals, medicating, surgery procedures, ear tagging, milking, birth assistance and others [41] which are often taught in education institutions such as animal science or veterinary courses. These HALTER and PULL_N results highlight the importance of habituating animals in educational institutions herd used in skill practice activities. Knowing that tactile stimulation is frequently used as strategy to desensitize the animal to human touching specially in dairy cattle [2,38,42,43] where the interaction between human and animal is very close and with constant tactile stimuli, this could also be used as a method to habituate animals in teaching herds before classes start. Providing a safer environment for both students and animals.

## 5. Conclusions

The current study shows that habituation of beef cattle in a teaching herd improves some animal behavior and physiology responses to human-animal interactions. In general, both habituation protocols showed benefits, but heifers that received the positive tactile stimulation in the chute showed the greatest improvement for chute enter and exit behavior. Furthermore, these heifers responded more calmly during student-animal interactions in class, which is beneficial for the students’ safety. To our knowledge, this is the first study that established beneficial effects habituation for cattle in an educational setting, in relation to student-animal interactions. More research is needed to optimize student learning whilst ensuring animal welfare.

## Figures and Tables

**Figure 1 animals-11-02159-f001:**
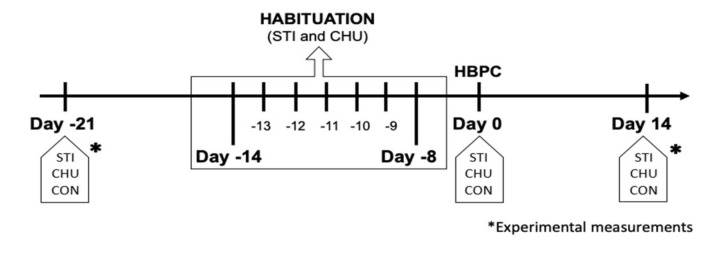
Timeline of the experiment activities at the Beef Cattle Center, relative to the start of the halter breaking practice class (HBPC). The habituation occurred on seven consecutive days Day −14 to −8) for the chute (CHU) and stimulated (STI) group. The Control group (CON) did not receive any habituation or handling prior to HBPC.

**Figure 2 animals-11-02159-f002:**
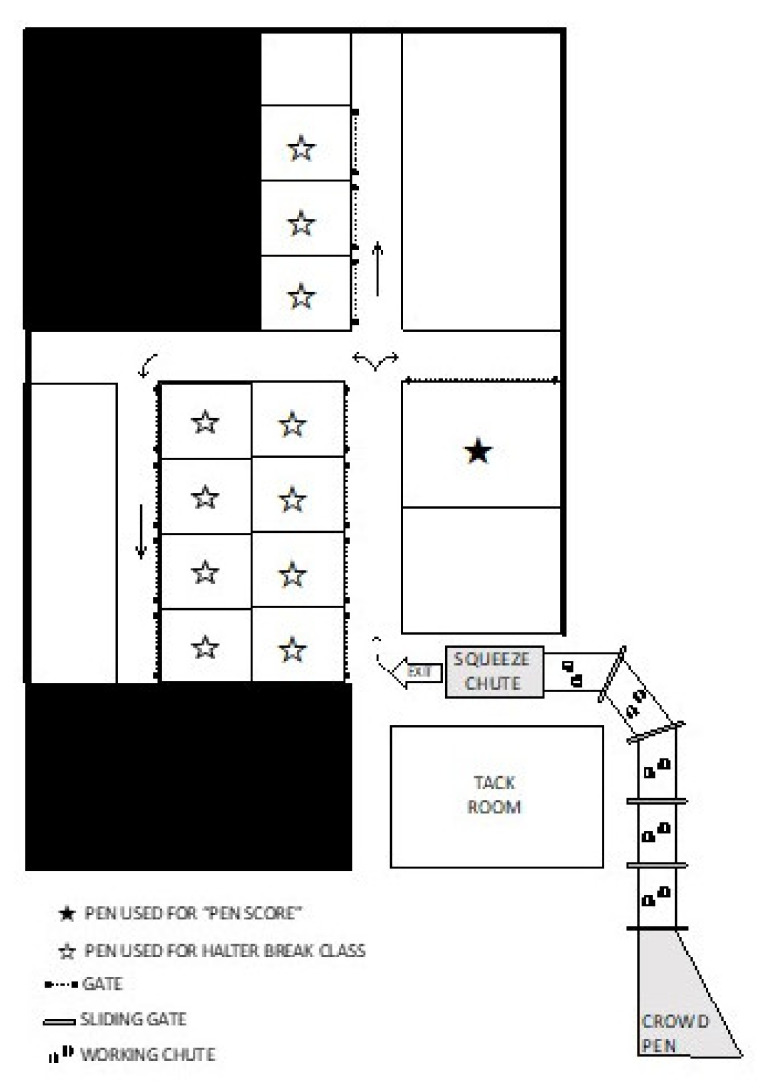
Corral layout at the university’s Beef Cattle Center. The arrows represent the direction that heifers were moved in, from the squeeze chute to the box stall used for the pen score (each ★ or ☆ represents one box stall). Cattle received tactile stimulation in the first 4 working chutes.

**Figure 3 animals-11-02159-f003:**
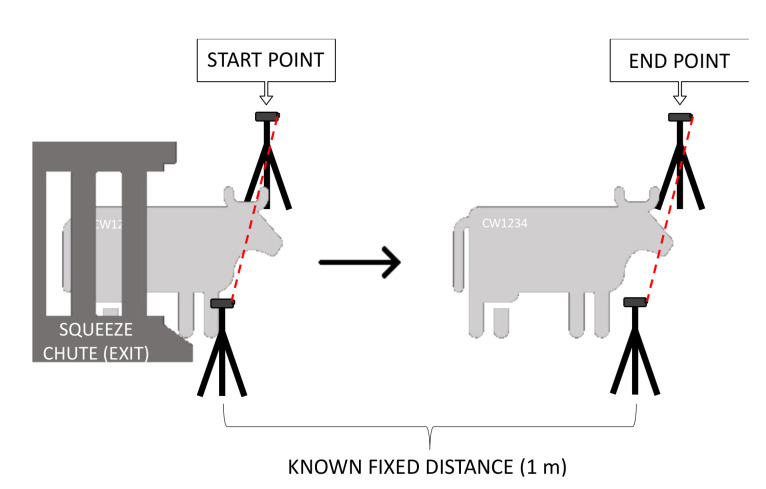
Flight time measurement. The sensors were placed on tripods, one sensor in each side (left and right) of the exit the way that they are parallel and facing each other. One pair of sensors (start point) were placed right after the headgate of squeeze chute (exit). The second pair (end point) were placed 1m in front of the first pair of sensors.

**Figure 4 animals-11-02159-f004:**
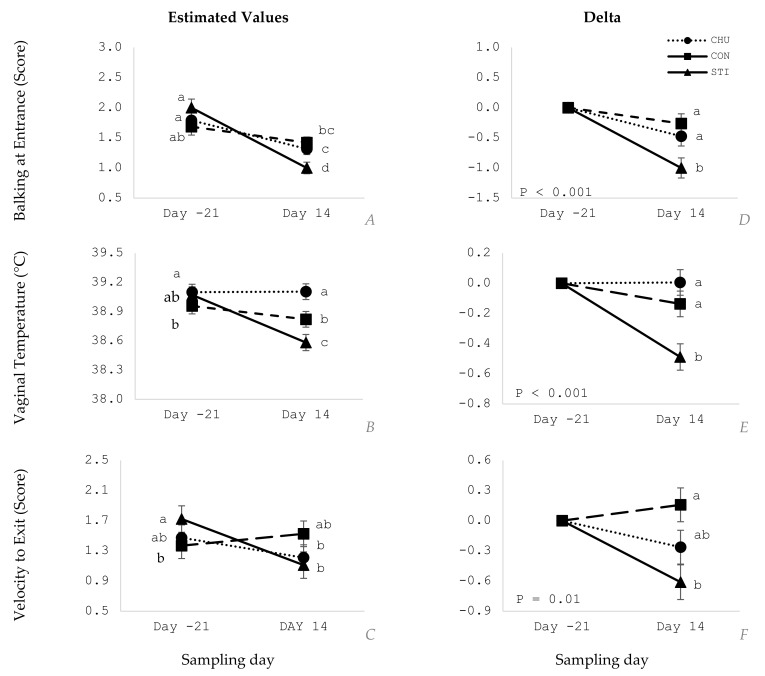
Estimated values (left column; **A**–**C**) with interaction effect of treatment and sampling day and, delta values (right column; **D**–**F**) with treatment effect of Stimulated (STI), Chute (CHU) and Control (CON) groups on sampling days −21 and 14. a, b: different lowercase letters indicate statistical difference (*p* ≤ 0.05) and equal letters do not differ significantly (*p* > 0.05).

**Table 1 animals-11-02159-t001:** Animal’s behavioral measurements from the chute entrance to chute exit measured on days −21 and 14 in the chute facility.

Moment	Parameter	Score	Animal’s Behavior Description	Reference
Working chute entry	Balk (BALK_ENT)	1	Entered voluntarily when the gate opens	Adapted from [2,19]
		2	Entered with a voice command or light tap on the rump	
		3	A hard slap on the rump or tail twisting was required to induce the animal to enter	
		4	The animal did not enter with the techniques described above	
	Velocity (VEL_ENT)	1	Walking—Defined as a four-beat gait characterized by progression of the alternate lateral legs, i.e., each hoof takes off from and strikes the ground independently	[20]
		2	Trotting—Defined as a two-beat diagonal gait in which the legs move in diagonal pairs, but not quite simultaneously	
		3	Running—Defined as a three-beat gait in which the front hooves strike the ground, one after the other in a fast manner, followed by the rear hooves	
	Latency (LAT_ENT)	*	Time to enter the chute, having the body completely inside the chute	Adapted from [21]
From working chute to exit	Locomotion (LOC)	1	Calm (quiet tail, eyes and ears), walked slowly without problem or balking	Adapted from [19,22]
		2	Calm (quiet tail, eyes and ears), walked slowly and balks sometimes	
		3	Calm (quiet tail, eyes and ears), walked fast without problem or balking	
		4	Calm (quiet tail, eyes and ears), walked fast and balks sometimes	
		5	Excited (restless tail, eyes and ears), walked slowly without problem or balking	
		6	Excited (restless tail, eyes and ears), walked slowly and balks sometimes	
		7	Excited (restless tail, eyes and ears), walked fast without problem or balking	
		8	Excited (restless tail, eyes and ears), walked fast and balks sometimes	
During blood sampling in squeeze chute **	Restraint (BC)	1	Animal remained calm, with no movement of body or hooves, during blood sampling	[20]
		2	Animal moved its head and body gently during blood sampling	
		3	Animal moved a lot, including movement of the hooves, and gently displaced the body	
		4	Animal moved briskly, shaking the body a lot, and moving the hooves simultaneously	
		5	Was struggling to escape—Animal was panicky or infuriated and struggles all the time to escape from the restraint during blood sampling	
Squeeze chute exit	Balk (BALK_EXT)	1	Exited voluntarily when the gate opens	Adapted from [2,19]
		2	Exited with a voice command or light tap on the rump	
		3	A hard slap on the rump or tail twisting was required to induce the animal to exit	
		4	The animal did not exit with the techniques described above	
	Velocity (VEL_EXT)	1	Walking—Defined as a four-beat gait characterized by progression of the alternate lateral legs, i.e., each hoof takes off from and strikes the ground independently	[20]
		2	Trotting—Defined as a two-beat diagonal gait in which the legs move in diagonal pairs, but not quite simultaneously	
		3	Running—Defined as a three-beat gait in which the front hooves strike the ground, one after the other in a fast manner, followed by the rear hooves	
	Flight time(FT)	***	Time taken to cut two pairs of electric eyes recorded electronically in the chute exit	[16]

* Was from the videos recorded during the experiment; ** Evaluated at the time of blood collection; *** Was measured in real-time by infrared sensors.

**Table 2 animals-11-02159-t002:** Behavioral measurements of the animals during the human-animal interaction in handling on the first day of halter breaking habituation.

Human-Animal Interaction Description	Parameter	Score	Animal’s Behavior Description
Student haltered the heifer in the squeeze chute	Reaction to haltering	1	Calm, easy to halter the animal
		2	Excited (alert), but still, being easy to halter the animal
		3	Excited (alert), moving the head gently away, but easy to halter
		4	Excited (alert), moving the head gently away, but difficult to halter
		5	Excited (alert), moving the head abruptly away, but easy to halter
		6	Excited (alert), moving the head abruptly away, but difficult to halter
Student conducted the animal from the squeeze chute to the pen	Velocity when leaving the squeeze chute	1	Walking calmly
		2	Walking excited or fast
		3	Trotting or running
	Velocity when entered the pen	1	Walking calmly
		2	Walking excited or fast
		3	Trotting or running
Student tied the halter’s rope of the animal to the pen’s fence	Reaction during first 5 min after tied	1	Standing still
		2	Pulling few times for the first seconds
		3	Pulling few times for a long time, but pulled less than 5 times
		4	Pulling and was struggling to escape all the time
		5	Pulling and was struggling to escape all the time; laying or turning the neck but not necessary to untie
		6	Pulling and was struggling to escape all the time; laying or turning the neck being necessary to untie
	Pulling behavior	-	How many times the animal pulled the rope during five minutes observation
Student started the tactile stimulation with the show stick	Reaction during two minutes when scratching started	1	Calm
		2	Excited (wagging the tail and/or moving the body) during approach, but soon calmed down
		3	Excited, wagging the tail all the time, but standing still
		4	Excited, wagging the tail, displacing the body once or twice
		5	Excited, wagging the tail, displacing the body all the time
Student conducted the animal from the box stall to the pasture field	Velocity when left the pen	1	Walking calmly
		2	Walking exited or fast
		3	Trotting or running

**Table 3 animals-11-02159-t003:** Pen score methodology adapted from [17].

Score	Animal Behavior Description
1	Very docile, approached the observer by itself (measured in the first minute after entering the pen).
2	Docile, did not react to the observer movements and allowed him/she to approach, even touch.
3	Docile, did not react to the observer movements and allowed him/she to approach but not to touch.
4	Slightly flighty, was aware of the observer and likely stand in a corner away from the observer.
5	Moved away from the observer and run with a raised head alongside the fence, fully aware of the observer’s position.
6	Flighty and was aware of the observer, may run along the fence or even run into gates or fences.
7	Very flighty, often called ‘crazy’ and often run at gates, fences and humans in an attempt to exit the pen.

**Table 4 animals-11-02159-t004:** Effects of treatment, day of sampling and their interaction on behavioral and physiological parameters of *Bos taurus* beef cattle (*n* = 56).

Trait	Unit	STI (*n* = 18)	CHU (*n* = 19)	CON (*n* = 19)	DAY −21 (*n* = 56)	DAY 14 (*n* = 56)	*p*-Value		
							TRT	DAY	TRT ∗ DAY
BALK_ENT	Score	1.50 ± 0.09	1.55 ± 0.08	1.55 ± 0.08	1.82 ± 0.08	1.25 ± 0.05	0.88	<0.0001	0.01
LAT_ENT	Seconds	2.03 ± 0.26 ^a^	2.99 ± 0.24 ^b^	1.92 ± 0.26 ^a^	2.00 ± 0.18	2.63 ± 0.24	0.01	0.04	0.24
VEL_ENT	Score	1.06 ± 0.07	1.13 ± 0.07	1.16 ± 0.07	1.23 ± 0.05	1.00 ± 0.05	0.54	<0.01	0.54
LOC	Score	4.69 ± 0.33	4.71 ± 0.32	5.45 ± 0.32	4.69 ± 0.27	5.21 ± 0.27	0.18	0.17	0.13
RR	Respiratory cycles/minute	44.00 ± 1.53	43.80 ± 1.51	46.63 ± 1.49	47.02 ± 1.24	42.60 ± 1.23	0.34	0.01	0.35
VAGT	°C	38.83 ± 0.06 ^a^	39.10 ± 0.06 ^b^	38.89 ± 0.06 ^a^	39.04 ± 0.05	38.84 ± 0.05	<0.01	<0.01	0.01
BC	Score	1.72 ± 0.16 ^a^	1.24 ± 0.16 ^b^	1.87 ± 0.16 ^a^	1.93 ± 0.17	1.29 ± 0.07	0.02	<0.01	0.62
CORT	µg/dL	1.94 ± 0.20	1.94 ± 0.19	2.34 ± 0.19	2.63 ± 0.16	1.52 ± 0.10	0.25	<0.0001	0.37
BALK_EXT	Score	1.19 ± 0.10	1.21 ± 0.10	1.18 ± 0.10	1.32 ± 0.10	1.07 ± 0.03	0.98	0.02	0.64
VEL_EXT	Score	1.42 ± 0.15	1.34 ± 0.15	1.45 ± 0.15	1.52 ± 0.10	1.28 ± 0.10	0.88	0.02	0.01
FT	Seconds/meter	1.42 ± 0.13	1.84 ± 0.14	1.58 ± 0.14	1.52 ± 0.11	1.71 ± 0.11	0.08	0.25	0.27
PS	Score	4.11 ± 0.17	4.24 ± 0.16	4.34 ± 0.16	4.57 ± 0.14	3.89 ± 0.14	0.62	<0.01	0.21

LSMeans ± standard error and *p* value. BALK_ENT (score): animal’s balking behavior when entering the working chute; LAT_ENT (seconds): latency to enter completely inside the working chute (starting the timer when the animal’s nose passed one point and ending the timer when the animal’s body passed that point completely); VEL_ENT (score): animal’s walking velocity score to enter the working chute; LOC (score): animal’s locomotion score from the working chute to squeeze chute; RR: number of respiratory cycle during one minute; VAGT (°C): vaginal temperature; BC (score): animal’s reactivity score during blood sampling; CORT (μg/dL): cortisol concentration in blood plasma; VEL_EXT (score): animal’s walking velocity score to exit the squeeze chute; FT (s/m): animal’s flight speed to exit the squeeze chute; PS (score): animal’s reactivity score in human presence and his approach inside the pen; STI: stimulated group; CHU: chute group; CON: control group; Day −21: 21 days before the halter breaking practice class and 7 days before habituation; Day 14: 14 days after the halter breaking practice class and 21 days after the habituation. TRT: treatment; DAY: day of sampling. ^a, b^ letters in the rows indicate statistical difference (*p* ≤ 0.05) and equal letters do not differ significantly (*p* > 0.05).

**Table 5 animals-11-02159-t005:** Effect of habituation treatment on behavioral parameters during the interaction between the students and their respective heifer (Bos taurus) in the halter break practice class.

Trait	STI	CHU	CON	*p*-Value
	(*n* = 16)	(*n* = 17)	(*n* = 16)	
HALTER	2.63 ± 0.34 ^b^	3.50 ± 0.36 ^ab^	4.38 ± 0.34 ^a^	*p* < 0.01
CH_EXT_VEL	1.81 ± 0.18	1.88 ± 0.18	2.00 ± 0.18	0.76
PEN_ENT_VEL	1.81 ± 0.21	1.88 ± 0.21	1.88 ± 0.21	0.97
PULL_N	8.13 ± 3.64	17.24 ± 3.53	16.31 ± 3.64	0.16
FIVEMIN	3.13 ± 0.25	3.24 ± 0.24	3.00 ± 0.25	0.80
SCRATCH	2.75 ± 0.32	2.82 ± 0.31	3.63 ± 0.32	0.11
PEN_EXT_VEL	1.69 ± 0.18 ^a^	1.80 ± 0.18 ^a^	1.13 ± 0.18 ^b^	0.02

LSMeans ± standard error and *p* value. HALTER (score): animal’s reactivity during the haltering; CH_EXT_VEL (score): animal’s walking velocity score to exit the squeeze chute; PEN_ENT_VEL (score): animal’s walking velocity score to enter the pen; PULL_N (n): number of times that the animal pulled the rope during the first 5 min after being tied; FIVEMIN: animal’s reactivity during the first 5 min after being tied; SCRATCH (score): animal’s reactivity during scratching with show stick; PEN_EXT_VEL (score): animal’s walking velocity score to exit the pen. a, b: different lowercase letters in the line indicate statistical difference (*p* ≤ 0.05) and equal letters do not differ significantly (*p* > 0.05).

## Data Availability

The data presented in this study are available on request from the corresponding author. The data are not publicly available due to the presence of identifiable information from human subjects.

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
