# Peer review of "Habituation Protocols Improve Behavioral and Physiological Responses of Beef Cattle Exposed to Students in an Animal Handling Class"

_animals, 2021, doi:10.3390/ani11082159_

Round 1
Reviewer 1 Report
Excellent paper. Great writing and data presentation.
I found it so enjoyable and I don't find reviewing many times to be enjoyable.
Author Response
Thank you very much for the review and kind words. We are excited to hear you enjoyed the manuscript.
Reviewer 2 Report
General comments
This study aimed to determine the impact of 2 different habituation protocols (movement through the chute with and without tactile stimulation) on the behavioural and physiological responses of beef cattle to handling. It also aimed to determine the impact of these responses on behaviour during a student halter-break class, with the aim of improving human-animal interactions to better safeguard human and animal safety. This study answers a useful question and takes a novel angle on a well-established body of literature around cattle habituation. The study is well designed, although the interpretation in the discussion is not always supported by the results presented. After revision, this study should provide a valuable contribution to the literature.
Summary
The specific summary is nice and clear but the abstract lacks the context of your study focusing on students handling a teaching herd. This should be clearer in the abstract.
Introduction
Overall a fairly good introduction to the study. Some parts of the introduction are a little repetitive and could be refined. It may also be useful to include more background on the halter breaking class and the standard procedures for such a class.
Methods
Overall it appears the methods used were appropriate to the question being answered. However, some of the methods are unclear and it is at times difficult to follow what happened and when. I suggest figure 2 could be expanded to include more detail of the timeline of behavioural and physiological measures. It would also be useful to restructure some of the methods and to present Figure 2 before Figure 1 to give a broader overview of the experimental design before going into details of the facilities.
I also have concerns about some of the behavioural scores being considered as continuous variables – for example reaction to haltering – does a score of 5 (head abruptly moving away but easy to halter) necessarily indicate increased reactivity compared to a score of 4 (head moving gently away but difficult to halter), or are these behavioural responses just different ways of coping at the same level of reactivity? Same for locomotion score – should excited and slowly walking without balking be scored higher than calm but walking fast and balking? Perhaps balking is a separate metric to movement speed/excitement. At the very least these limitations might be considered within the discussion.
Results
No major comments on results – see specific comments
Discussion
The discussion requires considerable revision. A lot of the discussion is unclear and difficult to interpret and the English needs to be improved. There are also a number of claims made at the moment are not supported by the results presented.
References
Not all of the references listed are referred to in the manuscript. The references are not listed in numerical order in some places (i.e. first paragraph of intro starts at [7] rather than [1])
Specific comments
Title: an active title is often nicer than a passive title e.g. Habituation protocols improved behavioural and physiological responses of beef cattle to student handling
21: physiological stress responses?
23: please specify - more positive behaviours compared to what?
23-25: suggest reordering the presentation of “training humans” and “strategies to habituate” in this sentence, to put the emphasis on the aspect of your study and better link to the previous sentence
34: change significant to significantly and clarify latencies to what?
36: 14 days after treatment or 14 days after the halter break class?
37: specify for all groups
39: first mention of student-animal interactions – introduce context of the halter break class sooner
Introduction
46: that reinforce (remove ‘s’)
48: may avoid prevent the animal to associate from associating humans
49-52: These sentences are a little unclear, please rephrase
61: cattle, may be distressing and difficult, due to students’ lack of experience, and of knowledge
62: Specify – ‘infrequent exposure to human-animal interactions’ of humans, cattle or both?
65: [3][4,5] –-> [3-5]
78: specify human interaction during a student halter break class
Methods
The methods could do with some restructuring to make it easier to understand the experimental design
86: specify if all these breeds were used in the study?
89: please specify - was there no interaction during this study period only or during their life to date?
91: I suggest you start by introducing the animal details here (total number, age, weight etc). Then explain the general experimental design and timeline before going into the details of the treatments. This can largely be achieved by swapping the order of the first 2 paragraphs and removing some repetition within the paragraphs. You can also switch the order of Figures 1 and 2 and add some additional details to the timeline of experimental activities. The timeline currently does not include the day 14 behavioural and physiological measures.
101: are were summed
102: class was were randomly assigned to one of three treatment
106: Table 1?
Figure 1: Some of the open star symbols are misaligned. You could indicate which of the 5 sections of the working chute contained the 4 animals being treated?
Figure 2: You can specify day 0 at the time of HBPC and extend to include the day 14 assessment (presumably 14 days after HBPC).
123: tactical à tactile?
124: Were the long handle, short handle and hands all used one after the other for equal durations in each session? Please specify here
130-131: I find this statement confusing – would the cattle not be moved individually through the whole setup as both the working chute and squeeze chute are single file?
147-150: This sentence is unclear, please rephrase. It may help to remove the text “resulting in 49 heifers included in the class”
151: handling were recorded collated from videos footage recorded taken with
153: Remove “Therefore”
157: via the working chute?
178: Please include in this paragraph the exact timing of blood sampling relative to the other procedures being conducted. Plasma cortisol concentration has a fairly acute peak after stress so timing is important
187: You can specify here that the behaviours are defined in Table 1
189, 190, 191: Please be specific about whether “chute” refers to the squeeze vs working chute throughout
189: The text “Behaviour on chute entry as balking” is unclear, please rephrase
189: perhaps refer to velocity as an entry or exit speed score instead – velocity implies you have recorded a speed (i.e. in m/s) which was not the case
192: I did not find BL to be an intuitive acronym, perhaps you could consider a different one?
194: Flight speed is a measure of m/s while flight time is a measure of seconds (and in this case seconds/meter). You are therefore reporting flight times throughout rather than flight speed – please correct the wording throughout accordingly (including in figures and tables)
197: Was latency to exit calculated like this too? There is no further mention of LAT_EXT after line 192 – did you measure this? Please either report these results or remove LAT_EXT from the methods
204: 1m seems like a small distance for flight speed assessment. The reference you give used a distance of 1.8m – is there any particular reason for this change?
214: Were any P values adjustments made to account for multiple comparisons?
Table 1: It would be useful to add the acronyms for the behaviours to this table for easy reference
How did you account for differences in human pressure applied to cattle when calculating latency to enter the chute? Was there a set amount of time that elapsed before moving to the voice command, light tap on rump, hard slap and tail twist?
236: Was measured timing from the videos… Evaluated just when have at the time of blood…
246: Animal’s bBehavioural measurements of the animals
Table 3: Velocity when left leaving the squeeze chute, Walking exited excited or fast
Table 3: Velocity when entered entering the pen, Walking exited excited or fast
Table 3: Excited (wagging the tail and/or moving the body) in by/during? Approach
Table 3: Velocity when left leaving the pen, Walking exited excited or fast
Results
253: second “BALK_ENT” should be changed to “BALK_EXT”
254: BL was also significant according to table 4
256: RR and cortisol are listed within the behavioural parameters rather than physiological
259: You can remove the stating of numbers here and in the next line, and simply refer to Table 4 at the end of the sentence instead
271: which differed only from CON group
274: You can remove the numbers and refer to Table 4 instead here too
Table 4: It could be useful to emphasise significant P values with bold font for easier reading?
Table 4 LAT_ENT and VAGT: swap the letters a and b so that a is presented first (as for BL)
282: to enter à when entering
283: time spent latency to enter
294: letters in the rows indicate
Figure 4: Check the positioning of the letters on Day -21 Vaginal temp and exit velocity graphs
Figure 4: Indicate significance value on the Delta plot for Balking
303-304: box stall exit – change to pen for consistent terminology
Discussion
Extensive correction of English language and style is required throughout the discussion and many of the paragraphs are unclear or grammatically incorrect. I have not made extensive comments on grammar or wording style at this stage, but the authors should seek advice to improve this
320: which results specifically?
321: is the perception of easier to handle cattle based on results presented in the study or the general feeling of the researchers and students handling cattle? Please specify in the text
324: leading – was this leading through with the halter or rather moving by using the flight zones of cattle? If the latter perhaps replace word ‘leading’ with ‘moving’
326: I am unsure if the treatments can be labelled as positive or neutral per se, as any contact would have caused some level of stress to the animals and may not have been perceived as positive or neutral – perhaps rephrase
328-333: This paragraph is largely just restating results. My interpretation of this paragraph is that it is attempting to explain the relationship between latency to enter, baulking and velocity score whereby latency can increase both because of a decrease in velocity (desirable behaviour) or an increase in balking (less desirable behaviour). Would suggest simply stating something like this then moving straight into the next paragraph (combine paragraphs) rather than restate results
338-342: I would suggest there are more relevant papers which could be cited to support these claims, perhaps some of Temple Grandins work?
343: again, please be specific about which results show…
343: I wonder whether excitement and movement speed are 2 different variables that may have a non-linear relationship? Combining these two factors together in the LOC score would therefore be problematic. You could potentially break the scores down into 2 factors and analyse them separately instead. This might then better support your inference that the CON heifers were more excited
345-346: This sentence is unclear
346-353: I do not believe your results can substantiate the claim that no animals in your study exhibited a passive coping style. Vocals were not recorded and if anything I would suggest balking may be indicative of a passive response more so than an active response.
351: RR did not significantly differ between treatment groups, nor was there a significant TRT*DAY interaction, according to the results you have presented (P=0.34, not <0.05)
356: Remove word “Therefore”
364-365: unsure why there is a need to classify an animal as ‘low’ for the interpretation of your results
369: Another reason could be differences in sampling protocols and timing
395: Specify what type of temperament scores
400-406: This section can be refined to reduce the restating of results
407: a good alternative compared to what?
Conclusion
Nice clear conclusion
456: remove “later in life” as this implies an ongoing effect beyond 14 days, which was not tested in this study
Reviewer 3 Report
I recommend to the authors an extension discussion, deepening the part of the discussion about the relationship between stress - cortisol and behavior
Author Response
We appreciate the review and comment. Extensive edits were made in the manuscript. Please see attached.